# Role of Endoscopic Sinus Surgery and Dental Treatment in the Management of Odontogenic Sinusitis Due to Endodontic Disease and Oroantral Fistula

**DOI:** 10.3390/jcm10122712

**Published:** 2021-06-19

**Authors:** Anda Gâta, Corneliu Toader, Dan Valean, Veronica Elena Trombitaș, Silviu Albu

**Affiliations:** 1Department of Otorhinolaryngology, University of Medicine and Pharmacy ‘Iuliu Hațieganu’, 400349 Cluj Napoca, Romania; veronicatrombitas@gmail.com (V.E.T.); silviualbu63@gmail.com (S.A.); 2Clinic of Neurosurgery, National Institute of Neurology and Neurovascular Diseases, 41914 Bucharest, Romania; corneliutoader@gmail.com; 3County Clinical Emergency Hospital, 400000 Cluj Napoca, Romania; valean.d92@gmail.com

**Keywords:** odontogenic sinusitis, oroantral fistula, endoscopic sinus surgery, dental treatment

## Abstract

Background: Odontogenic sinusitis (ODS) is frequently encountered in ENT practice; however, there are no guidelines regarding its management. This study aims to analyse the results of endoscopic sinus surgery versus dental treatment in ODS. Additionally, we aim to demonstrate the benefit of associating endoscopic sinus surgery (ESS) to surgical closure of chronic oroantral fistulas (OAF) by comparing mean time to healing in patients who opted or not for concurrent ESS. Methods: Records of patients with ODS were reviewed. Group one consisted of patients with ODS caused by periapical pathology undergoing either endoscopic sinus surgery (ESS) or dental treatment. Resolution of ODS was considered treatment success and was compared between the two treatment strategies. Group two included patients with ODS and associated chronic oroantral communication. Time to healing was compared between patients undergoing OAF closure alone versus patients receiving associated ESS, using the Log-Rank test to correlate Kaplan–Meier curves. Results: 25 patients from a total of 45 in group one underwent dental treatment alone, and 20 opted for exclusive ESS treatment. The failure rate was 40% for patients treated with ESS compared to 4% (one patient) for dental treatment. ODS resolved in all patients in the second group, but the mean time to healing was half (10 days) when ESS was complementary to OAF closure. Conclusion: The present study represents the first estimator of the role ESS plays in OAF treatment. Nonetheless, it provides proof of the importance of first addressing dental problems in odontogenic sinusitis.

## 1. Introduction

Odontogenic sinusitis (ODS) represents maxillary sinusitis, with possible extension to other sinuses, caused by the spread of adjacent dental pathology or resulting from trauma and oral procedures [1]. The various odontogenic pathologies, such as periodontitis, endodontic disease, oroantral fistula and sinus foreign bodies, act as a Trojan horse by passing oral pathogens inside the maxillary sinus [2,3,4].

The incidence of ODS was considered to be approximately 10% [5] of all sinusitis cases, but recent reports indicate rates of up 25% and even 45% of cases with unilateral sinus disease [6,7]. Even though ODS represents a pathology frequently encountered by rhinologists, there are no protocols regarding its treatment other than recommendations from centres with a high incidence of cases. Ferguson and Longhini [8] reviewed 85 sinusitis guidelines, and 11 mentioned ODS, but none gave guidance regarding treatment of this frequent pathology. Moreover, the most recently updated European Position Paper on chronic rhinosinusitis and nasal polyps (EPOS 2020) [9] recognizes ODS as a possible cause of rhinosinusitis but does not provide management guidance. ODS differs from ‘rhinogenic’ sinusitis in terms of pathophysiology and microbiology, thus requiring a different therapeutic approach [3,10]. There has been a lot of debate in the dental and medical literature regarding the proper treatment, timing and sequence of interventions [1,8,11,12,13,14]. However, no consensus has been reached other than a strong recommendation for shared decision-making between ENT surgeon, patient and dental care provider [2]. Oroantral communications represent a pathological pathway for oral bacteria towards the sinus, causing odontogenic sinusitis. Oroantral fistulas (OAF) develop when the oroantral communication does not close and becomes epithelialized [15]. Treatment of OAFs is attributed to maxillofacial surgeons, however, due to the commonly associated ODS, a collaborative effort with the ENT surgeon is frequently required.

The purpose of the present study was to compare results of dental treatment versus endoscopic sinus surgery (ESS) as sole treatments for patients with ODS. An appendage of our study was the assessment of ESS role in the treatment of chronic OAF.

## 2. Materials and Methods

This study was approved by our Institutional Ethics Committee (115/15 April 2021). We retrospectively reviewed the medical charts of all patients diagnosed with odontogenic sinusitis by S.A. in the II department of Otolaryngology Cluj-Napoca between 2008 and 2018. Only patients with ODS caused by dental caries and periapical abscesses who did not respond to medical treatment were included. For a better understanding, we will refer to patients with ODS caused by dental disease as group 1. Exclusion criteria consisted of chronic rhinosinusitis with nasal polyps, sinonasal malignancy, previous sinus surgery, ODS associated with implants, trauma, oroantral communications and with sinus foreign bodies. The diagnosis of ODS was made by the unilaterality of sinusitis, endoscopic findings (purulent discharge/polyp in the middle meatus/bulging uncinate process—Figure 1 and Figure 2), CT or CBCT interpretation (Figure 3 and Figure 4), and in certain cases, a correlation between nasal and dental symptoms. Although all patients presented maxillary sinus opacifiation, Lund Mackay scores could not be assessed due to some presenting with CBCTs limited to the maxillary sinus. All patients were given medical treatment (amoxicillin-clavulanate 2 g daily for 14 days and nasal decongestants) and were directed toward a dentist of their choice to confirm the diagnosis. Patients who failed medical treatment were offered either ESS or dental treatment, based on a decision in collaboration with the patient and the dental care provider. Only patients who failed medical treatment and underwent either dental procedures or endoscopic sinus surgery were included. For primary dental treatment, either root canals, with or without periradicular surgery, or dental extraction was performed. ESS was performed under general anaesthesia and was tailored to the extent of the disease. Uncinectomy, middle meatal antrostomy and anterior ethmoidectomy were performed for all patients, and, in some instances, posterior ethmoidectomy, sphenoidotomy and frontal sinusotomy was associated. After surgical treatment, patients were prescribed antibiotics for seven days, and saline nasal washes for 30 days.

A secondary search was carried out for patients with ODS and associated OAF. We only included patients who underwent surgical closure of the fistula by the maxillofacial surgeon. Patients with OAF were offered either ESS concurrent with the OAF closure or ESS performed in the following one or two days. The decision to undergo ESS was taken by the patient. For a clearer exposure, we will refer to patients with OAF as group 2, subdivided into patients receiving concurrent ESS, or not. OAF closure was performed using Rehrmann’s buccal advancement flap or the buccal fat pad [15]. ESS consisted of uncinectomy, maxillary antrostomy and anterior ethmoidectomy. All patients were followed up weekly for the first month and monthly after, recording their symptoms and nasal endoscopy findings.

Moreover, patients with OAF were instructed to contact their ENT doctor by phone daily after surgery and were also consulted the day their symptoms receded. Treatment success was determined by resolution of subjective symptoms and absence of sinusitis signs on nasal endoscopy. Statistical analysis was performed using Kaplan–Meier curves to assess the clinical outcomes between the different treatment strategies used in each group of patients. The difference between Kaplan–Meier curves was assessed using Log-Rank test—Figure 5 and Figure 6. Recurrence of maxillary sinusitis symptoms, confirmed by ENT examination with nasal endoscopy, was considered failure of treatment.

## 3. Results

A total of 45 patients (comprising 24 females [53.3%] and 21 males [46.7%]) were diagnosed with unilateral odontogenic sinusitis caused by dental disease, and 31 patients (17 males [54.8%] and 14 females [41.2%]) presented ODS associated with OAF. The mean age in group one was 53.1 years (range 41–65) and 54.3 in group 2 (range 40–63). In group one, the most common presenting symptoms of ODS were unilateral purulent rhinorrhea (19 patients in D vs. 17 in ESS), nasal congestion (14 in D vs. 16 in ESS), dental pain (9 in D and 6 in ESS), facial pain and pressure (4 in both subgroups) and foul smell (5 in D vs. 4 in ESS). Mean follow-up for group 1 was 19.5 ± 23.4 weeks, and for group 2, 21.3 ± 27.6 weeks. In group one, 25 patients selected dental treatment and 20 patients selected ESS without dental treatment during the follow-up period. Symptoms and findings on nasal endoscopy were similar between the two treatment subgroups. No significant complications occurred with either treatment strategy. Among the dental treatment patients, one (4%) patient was considered a failure due to persistence of signs and symptoms of ODS and underwent ESS. For the ESS subgroup, eight (40%) of 20 patients failed treatment, with five patients presenting persistence or recurrence of symptoms in the first month and three patients at two and three months. For patients with treatment success, the mean time to resolution of ODS for dental treatment was 14 days (range 10.1–17.8), while for ESS, it was 18 days (range 12.1–23.8). Applying the Log-Rank test for comparison of survival rates indicated a statistically significant difference between the two treatments (*p* = 0.001).

In the second group, OAF was caused by dental extraction, and ODS was either present before the dental extraction or ODS developed due to the oroantral fistula. All 31 patients underwent surgical closure of the OAF by the maxillofacial surgeon, and 20 patients opted for complementary ESS. Closure of OAF and resolution of odontogenic sinusitis was obtained in all cases (success rate 100%). The mean time to healing for patients who received ESS was 10 days (range 8.9–11.1), compared to 20 days for patients who refused ESS (range 15.4–24.5). The Log-Rank test for comparing survival rates demonstrated a statistically significant difference (*p* = 0.001).

## 4. Discussion

The differential diagnosis for unilateral sinus disease is broad, yet odontogenic sinusitis appears to be more frequently involved than historically described [1,16] and has shown an increase in the last decade [17]. The rise in incidence could be partly explained by the emergence of CT and CBCT and the tendency to replace routine dental radiographs with more precise examinations. Between 55% and 86% of dental pathology incriminated in ODS may be overlooked on dental radiographs [13,18]. On the other hand, dental pathology is also frequently missed by radiologists on CT scans. Turfe et al. [7] showed a rate of 65% diagnosis of dental disease on radiology reports, and it was similar regardless of dental pathology. As a result, otolaryngologists should carefully examine maxillary dentition on imaging studies, especially in the setting of unilateral maxillary sinus disease [19]. ODS caused by dental caries and periapical pathology represent a distinct type of ODS and its management differs from sinusitis caused by dental treatments [20].

Nonetheless, management options include primary dental or ESS treatment, or an association of the two. Several authors advocate good results in treating ODS with endoscopic sinus surgery alone [1,13]. Craig et al. [1] conducted a prospective study on 37 patients with ODS undergoing either dental treatment or ESS. The author found that sinusitis did not recur in patients with primary ESS, and a faster resolution of symptoms and endoscopy findings was obtained compared to the dental treatment group. Moreover, seven patients presented periapical disease in the ESS group and did not receive subsequent dental treatment. This made the authors recommend ESS as first-line therapy for symptomatic ODS and associate dental treatment when necessary. In our study population, the failure rate for ESS was 40% when the dental nidus of infection was not addressed. On the other hand, we do admit that a success rate of 60% for primary ESS sparks controversies. Nonetheless, the follow-up period is potentially insufficient and, as stated by Craig et al., there may be certain reversible dental causes of ODS. We consider this finding intriguing and further studies to be worthwhile. Wang et al. [13] presented a retrospective analysis of 55 ODS patients provided with heterogeneous treatment options. Of these, 33% resolved with ESS alone, 33% with concurrent dental surgery and ESS, while only 10% resolved with dental surgery alone. Even so, they do consider it necessary to target the dental source of infection and do not recommend ESS as the first and only treatment option. The reason the results mentioned above differ significantly from ours could be explained by the fact that Wang et al. also included other dental pathologies but did not provide detail upon the selection of treatment in regard to the ODS aetiology. In addition, our service is joined with the emergency maxillofacial department, consequently, it is more likely that our patients presented with more advanced disease. Our paper has shown a statistically significant benefit of dental treatment first, with a failure rate of only 4%, this being in accordance with other similar case reports [8,14,21,22].

In 2018, the American Association of Endodontists published a position paper on sinusitis of endodontic source, which can be considered one of the first treatment guidelines for this pathology. This paper’s recommendations are per our study results since it advises physicians first to perform dental treatment, followed by ESS only if needed [23]. A more recent multidisciplinary consensus statement [2] acknowledges that for the patients with minimal sinonasal symptoms of ODS from treatable dental pathology, a primary dental treatment should be pursued.

Regarding the need for subsequent ESS, Mattos et al. [14] proposed high Lund–Mackay scores on the CT scan and involvement of the ostiomeatal complex as predictive factors for the necessity of undergoing ESS. This is in accordance with a more recent analysis by Yoo et al. [22], which also proposes early ESS for patients with smoking habits. Concerning the timing of the two treatments, the consensus released in 2020 [2] recommends considering ESS in patients with persistent sinusitis 1 to 2 months after dental treatment.

Our analysis of ODS treatment, however, does not assess the third option of concurrent dental and ESS treatment, with high reported success rates (90% to 100%) [2,12,20]. Nonetheless, we consider that performing both dental and ESS treatment is not always necessary since additional surgery is associated with higher risks, and many of the ODS produced by odontogenic infections resolve by dental treatment alone.

An alternative approach proposed by Albu et al. [24] is the association of endoscopic canine fossa puncture (CFP] at the time of dental treatment in ODS without OAF. This technique has proved to be an effective conservative treatment with several advantages: it can be performed under local anaesthesia and provides broad exposure of the entire sinus and maxillary ostium. We could recommend the CFP procedure for patients without predictive risk factors for requiring ESS by extrapolating the information mentioned above.

Felisati et al. [20] proposed a classification of ODS according to aetiology and named. Group III of ‘classic’ dental disease and treatment complications was further divided into classes according to the presence of the associated OAF. The same team from Milan found this group to be the most common cause of ODS, and 34.8% of these patients associated an OAF [10]. Immediate closure of acute oroantral communications had a success rate of almost 95%, but the success rate decreases with a secondary closure [25]. However, many authors describe better success rates when ESS is associated [26,27,28], and some [29,30] have even noticed a faster healing. Despite these observations, this is the first study to the best of our knowledge aiming to precisely quantify the difference in the healing time between simple OAF closure versus closure with concurrent ESS. Even though all patients in group 2 presented treatment success, Kaplan–Meier curves showed a significantly faster healing for patients undergoing concomitant ESS and dental treatment. Since it was a retrospective design, we could not evaluate which patients presented ODS before having the oroantral fistula; this, of itself, would be an interesting future investigation.

Our study has several limitations. First of all, retrospective collection of data can present bias in patient selection and does not provide access to all patient details. Additionally, some patients were referred from the dentist with CBCTs that only included the maxillary sinus, and we could not determine the extent of sinus disease. Consequently, Lund–Mackay scores could not be assessed, and we could not identify which patients were smokers. Also, we could not reexamine all CT/CBCT scans of patients with unilateral sinus disease and, as such, there can be patients in our service that have been overlooked. Moreover, in group one, dental examination and treatment were not performed by the same doctor, so data on the exact procedures are not available. Also, we could not establish the frequency of each dental care provider (general dentist, endodontist, periodontist, or oral surgeon). We consider further research to be necessary, with more details provided about dental procedures performed.

## 5. Conclusions

ENT surgeons should always consider ODS when faced with unilateral sinus disease, and treatment should be tailored to each patient. This study has reinforced the idea that addressing the dental cause is pivotal to treating this pathology. More importantly, a proper collaboration between dental care providers/maxillofacial surgeons and otolaryngologists has proved to be of utmost importance. Furthermore, we managed to provide a quantification regarding the exact value of ESS in speeding the recovery after OAF closure.

## Figures and Tables

**Figure 1 jcm-10-02712-f001:**
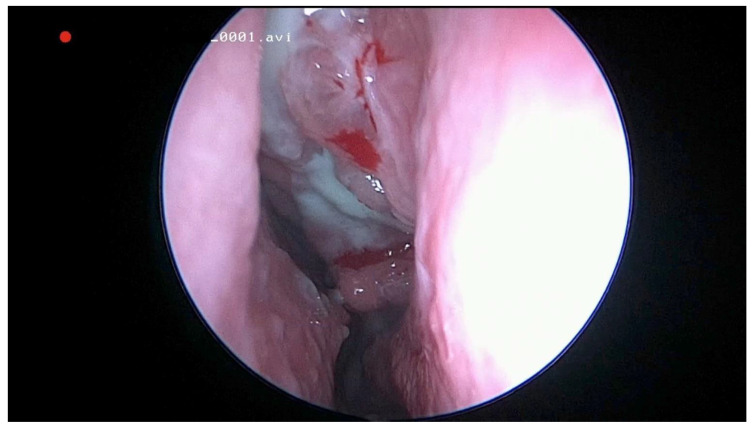
Endoscopic view of the left middle meatus with polypoid tissue and purulent drainage from the maxillary sinus in a patient with odontogenic sinusitis.

**Figure 2 jcm-10-02712-f002:**
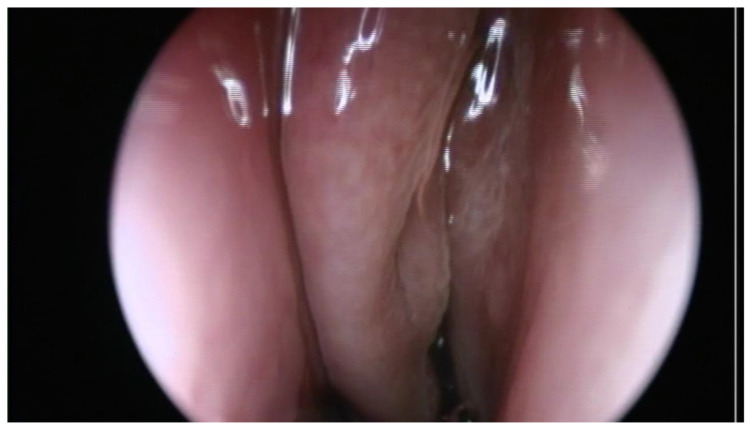
Endoscopic view of left middle meatus in a patient with odontogenic sinusitis showing a bulging uncinate.

**Figure 3 jcm-10-02712-f003:**
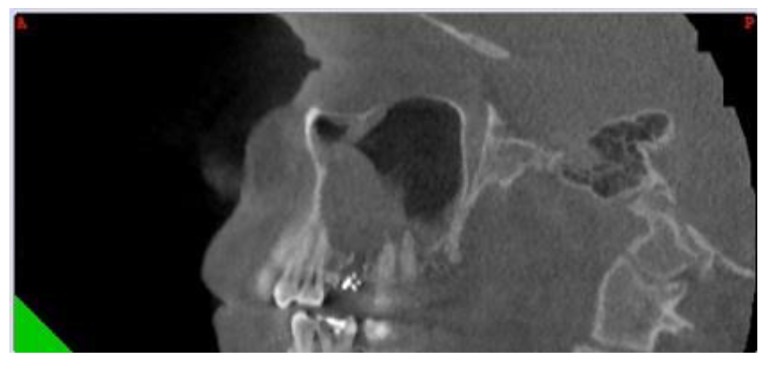
Sagittal CBCT of a patient with left odontogenic sinusitis and periapical fistula located at the level of the first molar, marked with a white asterix).

**Figure 4 jcm-10-02712-f004:**
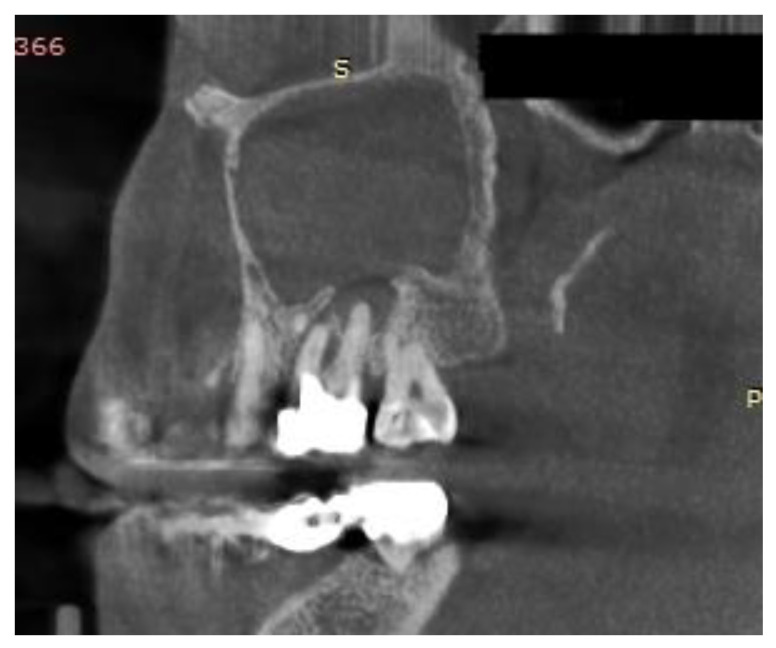
Sagittal CBCT of a patient with odontogenic sinusitis caused by a periapical abscess.

**Figure 5 jcm-10-02712-f005:**
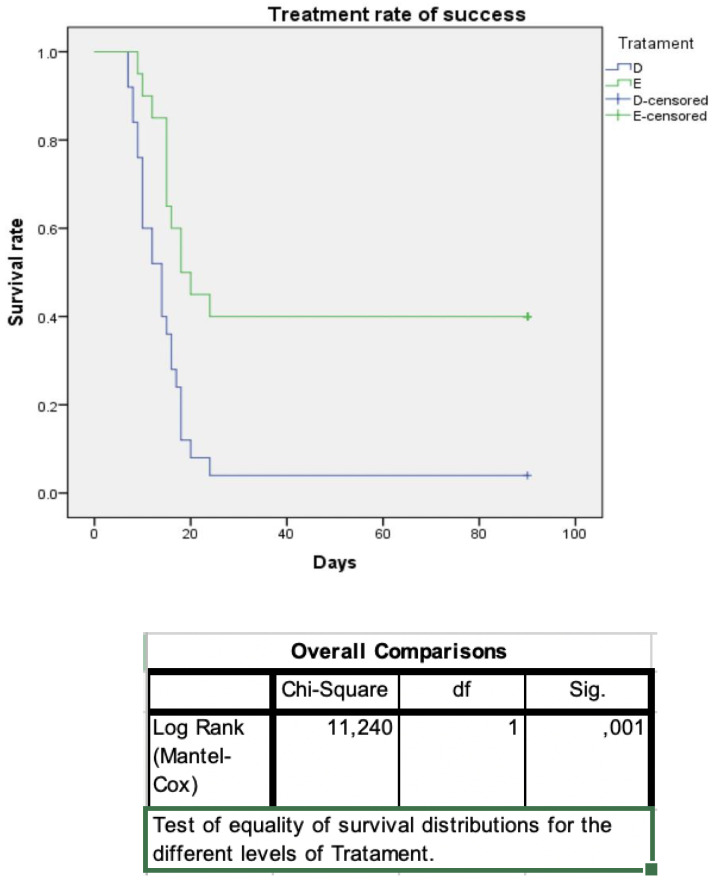
Kaplan–Meier curves showing healing time between patients receiving dental treatment (D) or endoscopic sinus surgery (E). 4% failure rate in D group and 40% in E group. Mean time to healing was 14 days for D, and 18 days for E. 1 Log-Rank test for comparing survival rates shows a statistically significant difference (*p* = 0.001) between dental end endoscopic treatment.

**Figure 6 jcm-10-02712-f006:**
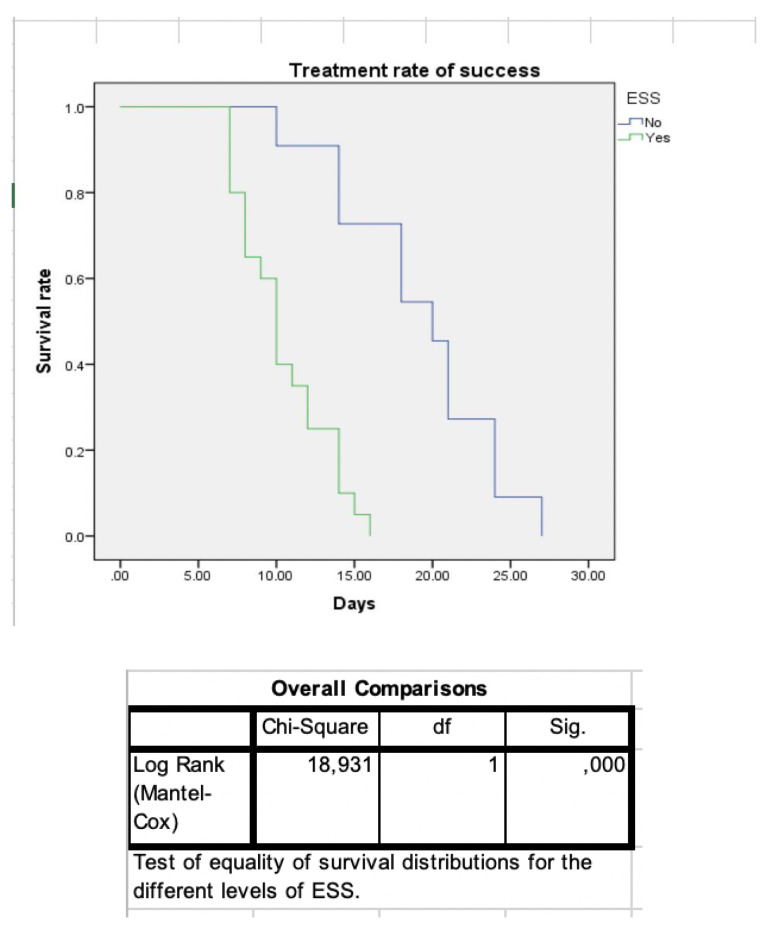
Kaplan–Meier curves representing time to resolution of odontogenic sinusitis between patients undergoing oroantral fistula alone or concomitant endoscopic sinus surgery (ESS). Mean time to healing was 10 days for ESS, and 20 days when ESS was not associated. 1 Log-Rank test for comparing survival rates shows a statistically significant difference (*p* = 0.001).

## Data Availability

The data presented in this study are available on request from the corresponding author. The data are not publicly available due to privacy reasons.

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
