# Peer review of "Role of Endoscopic Sinus Surgery and Dental Treatment in the Management of Odontogenic Sinusitis Due to Endodontic Disease and Oroantral Fistula"

_jcm, 2021, doi:10.3390/jcm10122712_

Round 1

Reviewer 1 Report

Indeed concurrent ESS in oroantral fistula treatment is an ongoing debate, common (physiologic) sense tells us its beneficial (ventilation), but evidence is weak. Studies like this are therefore needed.

A few issues.

The title sounds like its going to be a letter or a opinion rather than an actual study. "Controversies in the management of odontogenic sinusitis"

For an actual original article its rather short

Line 82 apparently some received ESS a week later. How many? How can there still be a quicker healing when ESS is performed later?

- retrospectively reviewed - but follow-up is extensive - was this intended, therefore a prospective design?? This should be ethically approved. "All patients were followed up weekly for the first 86 month and monthly after, recording their symptoms and nasal endoscopy findings."

Reviewer 2 Report

First, I congratulate the authors on a nice retrospective study. This is an important area of research to develop. I have reviewed a lot of papers on odontogenic sinusitis, and I appreciate your use of appropriate terminology and referencing of recent important literature. Below are points that I think will strengthen your paper.

Title:

- The title should be changed to reflect the findings of the study, which will increase the draw of the work. The current title makes it sound more like a review of current management controversies, but your study is an original research study and the study should capture this. For example, the title should read something along the lines of “roles of dental treatment and sinus surgery in the management of odontogenic sinusitis due to endodontic disease and oroantral fistula.”

Introduction:

Good. No comments

Methods:

- Could you specify the different dental provider types managing your patient, and can you determine frequencies of each? (general dentists, endodontists, periodontists, or oral surgeons)

- Periradicular surgery was definitely performed on some patients? This is normally performed after a failed root canal to my knowledge, so this would not be a primary dental treatment. Please clarify in paper how this was utilized, because readers may not know about this procedure.

- Can you determine degrees of sinus disease on CT, specifically did you include any patients with only maxillary sinus mucosal thickening?

- Why did you mention viewing the alveolar recess? If you did not use this data, you can delete. If this information was used, then please specify how.

- Typo possibly in last sentence of first methods paragraph: saline washes instead of washed?

- You followed all patients weekly for month, and monthly for how long? Please detail how long patients were followed. Were patients only followed until resolution?

               - You should also specify how many patients were lost during follow-up

- Please specify how recurrence was defined. Was it based on nasal endoscopy alone? Could patients have been considered recurrences with symptoms, but no infection seen on endoscopy?

- For OAF closure groups, you should state there were 2 groups: one who had complementary ESS (concurrent or 1 week after OAF closure), or no ESS. This will help make results clearer.

Results

- Is it possible to report specific frequencies of symptoms between dental and ESS treatment patients? Something more specific than “endoscopy and symptoms were similar between the 2 treatment groups”? If not, then technically we can’t know they were similar. You could mention in limitations that specific symptoms and frequencies were not analyzed, and future studies should explore whether certain symptoms are more or less likely to resolve with different treatment modalities.

- How long were dental and ESS patients followed (both for endodontic disease and OAF groups)? We need to know mean follow-up duration and standard deviation. Some patients can recur after presumed to have resolved.

- You wrote 8 patients failed ESS, but then reported 9 later in the sentence. Clarify.

- The mean time to resolution for the ESS group I presume was only for the 12 patients who resolved? Or did some of the 8 failures resolve, and then recur? And again important to how long were they followed for recurrence, and how many of these patients got dental treatment eventually? If you don’t have that data, please clarify why. The main concern of course is that this looks like 60% of patients resolved with ESS without any dental treatment. Should discuss this in discussion section

- Can you determine how many of the OAF patients had ODS before or after extraction? This would be publishable in and of itself, as it has yet to be published, and is very important to appreciate.

Discussion:

- I reread the paper you referenced by Craig et al., and I think you should delete the word “only” when summarizing their recommendations. They reported at different places in the paper that primary ESS could be recommended for symptomatic ODS, followed by dental treatment “when necessary” or “as needed.” I realize the difference is subtle, but the word “only” made it seem to me like that paper was somewhat against dental treatment. Also technically they didn’t use that word. And while you don’t have to mention this next point, they also stated that patients can always be offered primary dental treatment regardless of symptom status.

- You should also add a point to follow that similar to Craig et al., you too found that some patients seem to resolve with ESS alone, but that long-term follow-up was not reported, and some of these patients could still recur, and that further research is necessary to understand whether some ODS patients do not require dental treatment. Until then, however, it’s safe to say we should be recommending dental treatment for ODS due to treatable dental pathology.

Limitations that should be discussed in paper:

- No mention of extent of sinus disease (number of sinuses opacified, and whether sinuses had mucosal thickening or opacification)

- Total follow-up time not reported

- Did not specify frequencies of the different types of dental treatments would be a major limitation for sure

Author Response

Plase see attachment

Round 2

Reviewer 1 Report

better modified version, at least a bit longer

Author Response

Honored Professor,

Thank You very much, again, for reviewing our paper. We appreciate your input and have sent our paper for an English spelling verification at an authorised professor. We have made the necessary corrections regarding correct language usage. We herein upload the corrected version of the manuscript.

Thank You again and I remain sincerely yours.